# Rapid detection of fast innovation under the pressure of COVID-19

**Nicola Melluso**[1]☉, **Andrea Bonaccorsi**[1]☉*, **Filippo Chiarello**[1]‡, **Gualtiero Fantoni**[2]‡

**1** Department of Energy, Systems, Territory and Construction Engineering, School of Engineering, University of Pisa, Pisa, Italy, **2** Department of Civil and Industrial Engineering, School of Engineering, University of Pisa, Pisa, Italy

☉ These authors contributed equally to this work.
‡ These authors also contributed equally to this work
* andrea.bonaccorsi@unipi.it, a.bonaccorsi@gmail.com

**Data Availability Statement:** The data underlying the results presented in the study are available from https://www.kaggle.com/nmelluso/technology-coronavirus. The data underlying the

## Abstract

Covid-19 has rapidly redefined the agenda of technological research and development both for academics and practitioners. If the medical scientific publication system has promptly reacted to this new situation, other domains, particularly in new technologies, struggle to map what is happening in their contexts. The pandemic has created the need for a rapid detection of technological convergence phenomena, but at the same time it has made clear that this task is impossible on the basis of traditional patent and publication indicators. This paper presents a novel methodology to perform a rapid detection of the fast technological convergence phenomenon that is occurring under the pressure of the Covid-19 pandemic. The fast detection has been performed thanks to the use of a novel source: the online blogging platform Medium. We demonstrate that the hybrid structure of this social journalism platform allows a rapid detection of innovation phenomena, unlike other traditional sources. The technological convergence phenomenon has been modelled through a network-based approach, analysing the differences of networks computed during two time periods (pre and post COVID-19). The results led us to discuss the repurposing of technologies regarding "Remote Control", "Remote Working", "Health" and "Remote Learning".

## 1. Introduction

The current epidemic crisis, the most dramatic since last War, is challenging many organizations and technological communities to be in a state of urgency for innovation [1]. To quickly deploy innovative solutions to radically new problems and to perform successfully in a hyper-competitive and unstable environment, organizations must develop dynamic capabilities based on agility, flexibility and speed [2]. Thus, sustainable fast innovation is a strategic priority for every type of organization (business, government, or nonprofit enterprise) [3–5].

In a few months technological communities around the world have demonstrated a remarkable mobilization and generated creative solutions to the extreme requirements of the pandemic. In particular, it has been noted that the technological community has rapidly

figures are available as Supporting Information Files.

**Funding:** The authors received no specific funding for this work.

**Competing interests:** The authors have declared that no competing interests exist.

redefined the agenda of research and development. Von Krogh [6] have noted the "ultrafast approach to innovation centered on the repurposing of readily available ideas, knowledge and technologies".

The destructive current situation brought by the global pandemic of COVID-19 is an extreme example of changes that are occurring at unprecedented rates of velocity and scale [7,8]. This has forced organizations to rethink their approach to innovation, defined as the "deployment of new ideas and/or technologies to create new or additional value" [9,10]. In the current landscape a sustainable model of innovation is based on the fast convergence of seemingly heterogeneous and unrelated things into solutions that can create exponential outcomes [1,11,12]. This concept is labeled convergence innovation (CI) and exploits network effects and the exponential economies of convergence [1].

In this paper we offer quantitative evidence of the fast innovation approach followed by public and private technologists under the pressure of the Covid-19 pandemic. We contribute to the literature in three ways.

First, we stress the need for fast detection of changes in technology. The Covid-19 crisis has demonstrated the need for intelligence approaches that take place in real time with the changes in underlying technological ideas, knowledge and solutions. The large literature on emerging technologies has developed several sophisticated methods for the detection of trends, hot topics, and weak signals, but the data used, i.e. patents and publications, provide inevitably lagged responses. These sources are valid and reliable, but suffer from a long latency. Under extreme pressure this limitation is fatal. We suggest a methodology for the exploitation of a recently developed repository of scientific and technological content that is feeded daily with a very short submission procedure, i.e. Medium. Not only Medium offers original content in the thousands, but it asks writers to actively tag the articles. Tags are therefore similar to keywords and convey a precise understanding of the content of papers.

Second, in order to detect new technological trends through the use of technical documentation (e.g. scientific papers and patents) a well known problem lies in the definition of query to retrieve relevant documents. It is well known that expert-based queries suffer from a number of limitations [13]. To overcome these limitations a dynamic, real time approach to delineation based on the hyperlinks of Wikipedia has been suggested [14,15]. This approach allows the continuous updating of the content of emerging technologies, exploiting the graph-like properties of the pages of Wikipedia. For this reason, we use the last version of Industry 4.0 delineation based on Wikipedia [14] in order to filter the Medium papers that deal with technologies directly or indirectly related to this paradigm.

Third, we contribute to the substantive literature on technological convergence and the emerging literature on repurposing and fast innovation. The literature on convergence emphasizes the long time spans and the obstacles to integration stemming from absorptive capacity, path dependency in learning, need for proximity and relatedness in knowledge. We give evidence that these obstacles can be overcome under severe pressure.

The present paper is structured as follows. Section 2 discusses the academic literature concerning the technological innovation occurred during the Covid-19 crisis. Section 3 reviews the theoretical work on the innovation repurposing and technological convergence. Section 4 presents data sources used to perform the fast detection of technological convergence. Section 5 describes the conceptualization of the process and techniques adopted for this work. Section 6 presents the results focusing on the two time frames before and after the pandemic analysing the differences according to our conceptualization. Section 7 discusses the phenomena detected with this experiment and concludes.

## 2. Innovation, technology and coronavirus: State of the art

The coronavirus pandemic is stimulating extraordinary innovation ideas. As damaging and tragic COVID-19 has been, in both depth and scale, one fortunate thing is that the pandemic occurred in a digital age [8]. The negative effects would have been much worse if we were not living in a time of advanced digital technologies and science [1].

In the scenario of the Fourth Industrial Revolution and digital transformation, the innovation is based on convergence of advanced technologies and strategic ideas [1]. Advances in digital technologies occur at a high speed, such as cloud-based ubiquitous computing, big data analytics, artificial intelligence (AI), machine learning, Internet of Things (IoT), autonomous systems, smart robots, 3-D printing, and virtual and augmented reality (VR & AR). For example, these technologies have been used in innovative solutions to manage the pandemic through real-time scanning of the virus spread, data analytics for testing, contact tracing, and isolation of infected patients [16].

As scientific research is moving towards focusing on solutions to manage this pandemic, we can list some examples of innovation generated at an accelerated speed [8,17].

First, a stream of innovations comes from the repurposing of products, facilities and drugs. Several manufacturing firms shifted their innovation efforts with the shared singular purpose of fighting the virus. In particular, these organizations were forced to rapidly change their operations and focus on producing medical equipment. For example, the conversion of manufacturing facilities or the repurposing of assembly operations leads to producing face shields and ventilators [18,19].

Second, resilient and agile engineering and scientific solutions have been pursued to address the societal challenge of the coronavirus pandemic with medical and manufacturing responses. These include advances in instrumentation, sensing, use of lasers, fumigation chambers and development of novel tools such as lab-on-the-chip using combinatorial additive and subtractive manufacturing techniques and use of molecular modelling and molecular docking in drug and vaccine discovery [20,21].

Third, digital technologies, such as artificial intelligence and the internet of things, played an important role for accelerating the process of digital transformation [8]. For example, cloud computing contributed to the shift towards people-less companies (at least partially) and touchless customer services. Many companies are transforming their operations by removing human touchpoints, toward a robotic leap forward. Online businesses, which have thrived during the pandemic, using algorithms and automation save cost, boost efficiency and protect workers and public health [22].

However, the business environment is turbulent and in constant change. The destructive current situation brought by the global pandemic of COVID-19 is affecting health care, economy, transportation, and other fields in different industries and regions [23]. A major negative impact is related to firm performances [24]. Today many firms, especially small and medium enterprises (SMEs), are struggling to find survival plans for the next quarter or next month [25,26]. To survive in this turbulent environment, organizations must be agile with dynamic capabilities. This innovation process requires not only collective intelligence to repurpose for shared goals, but also collaborative efforts to converge different ideas with speed and utmost determination [27].

Under these assumptions, Lee and Trimi [1] developed the concept of convergence innovation. They argue that innovation has evolved throughout history, going through different phases or stages: (i) closed innovation, (ii) collaborative innovation, (iii) open innovation (iv), co-innovation and (v) convergence innovation. They proposed the last phase as a sustainable innovation strategy, with its autonomous ecosystem, in the turbulent digital age. This new

concept can be a catalyst for managing the current COVID-19 pandemic and charting the path to post crisis by helping organizations to implement effective strategies for value creation with agility.

## 3. Innovation repurposing and technological convergence

The pandemic crisis has generated the need for fast solutions to new problems. Von Krogh [6] have argued that companies are adopting a repurposing approach. This approach has initially been invented by pharmaceutical companies to identify additional therapeutic uses of existing drugs. In the same work it is described a number of case studies of companies that, under the pressure of Covid-19, are using their technologies and manufacturing facilities to produce goods in high demand due to the sanitary emergency (e.g. ventilators, hand sanitizers, disinfectants, face shields, prefabricated hospitals) or developed new technologies to address the health crisis (e.g. AI-based diagnostic tools). In some cases discussed by the authors there has been a convergence between widely different knowledge bases, for example between engineers and physicians, or between Artificial Intelligence experts and medical doctors with CT expertise.

As a matter of fact, recent studies strongly suggest that technological convergence is on the rise [28–30]. In fields such as electronics, telecommunication, media, information technology, energy, mobility, mechatronics, as well as in some areas in health care, we witness a number of technologies that are generated from the convergence of previously separate areas. Convergence may originate in science (science convergence, mainly related to interdisciplinary and transdisciplinary research), in technology [31], in market demand [32]. The joint occurrence of technology and market convergence, or supply and demand, gives in turn origin to industry convergence [33].

From a strategic perspective, achieving convergence among heterogeneous technologies requires strong coordination. Several authors argue that this can be better achieved within the boundaries of firms, for example by using parent-subsidiaries relations [34], given the possibility of failures in inter-company cooperation [35]. At the same time, the recent literature on innovation ecosystems shows a number of cases in which convergence is achieved through horizontal and vertical cooperation between independent companies. Hwang [30] identified more than 9,000 collaborations over 35 years in Korea and found that inter-firm collaborations in ICT were the most effective for technological convergence, while collaborations with universities were less significant.

There is large agreement, however, on the idea that there are severe constraints on technological convergence. Heterogeneous technologies are difficult to master within the boundaries of a single company, due to the limits of absorptive capacity [36,37]. According to Caviggioli [38] convergence is more frequent if the focal technology fields are closely related, with a large number of cross-citations in patents. Duysters and Hagedoorn [39] found that the number of strategic alliances in neighbouring sectors increased more than average in the 1980–1993 period. Fai and Tunzelmann [40] found strong persistence over time of patent fields at company level by using Revealed Technological Advantage indicators. In a certain sense, these studies point to the possibility of a negative impact of external technology: the impact is inversely related to the technology distance [41].

According to Jeong and Lee [31], furthermore, there is a need for a long timespan of R&D projects in order to achieve convergence. The earlier the stage of Technology Readiness Level, the earlier the stage in the Technology Life Cycle and the longer the time span of collaboration in R&D, the larger the probability of convergence. Interestingly, the scale of investment is not relevant.

## 4. A source for fast detection: The medium blogging platform

The detection of technological convergence has been approached with studies that use scientific articles, patent data or a combination of the two [42–47]. Other studies have access to microdata on inter-firm collaboration and examine the structure of networks [48] or use interviews [49]. Convergence is measured by using various definitions of similarity, applied to existing classifications (Subject Categories of publications, IPC classes of patents), or to items in the text of documents (title, keywords, abstract, full text) [50].

The rapidity of spread of the coronavirus showed the need for a process of fast recognition of technological convergence. The use of traditional sources (publications, patents, or microdata on inter firm collaborations) suffers from latency in availability. For this reason, we must take into account sources that allow a rapid accessibility of information, such as online social networks platforms. During the pandemic, online social networks (e.g. Twitter, Medium, Facebook) have become the primary source of communication. In fact, due to the large user-base and their high pace of textual production, these online social networks are becoming the source of vast amount of data and establishing new research dimensions, such as social computing, predictive modeling, and big data analytics [51]. The volume and velocity of these platforms led us to the decision to tackle the problem of fast detection of technological convergence by offering an exploratory exercise on a blogging platform that produces knowledge on a daily basis, i.e. Medium.

Medium is one of the most trending blogging web-platforms. The platform is an example of social journalism, having a hybrid collection of amateur and professional people and publications, or exclusive blogs or publishers on Medium, and is regularly regarded as a blog host. Medium has many interesting characteristics (positive and negative) to discuss.

First, the platform presents a hybrid structure that accomplishes most of the features of the other social networks. Once a user posts an entry (called "story"), it can be recommended and shared by other people, in a similar manner to Twitter; posts can be upvoted in a similar manner to Reddit, and content can be assigned a specific theme, in the same way as Tumblr. These features ensure an easy adoption and growth for the platform, since they acted as trump cards for other social networks.

Second, posts on Medium are sorted by topic rather than by writer, unlike most blogging platforms (e.g. Blogger). The platform uses a system of "claps", similar to "likes" on Facebook, to upvote the best articles, called the "Tag System". In practice, at the moment of publication, authors are forced to feature an article with tags. Tags are words (or multi-words) that capture the essence of the article. Thanks to the "Tag System", the tags make the article searchable and ensure to get more reads. Hence, the tags are similar to the keywords for scientific publications. This feature makes possible the treatment of stories on Medium similarly to scientific articles. In fact, an advantage of this platform is the possibility to perform typical analysis based on keywords.

Third, Medium can work as a publishing platform with its internal framework of "Publications". "Publications" are shared spaces for stories written around a common theme or topic, usually by multiple authors. In practice, "Publications" are distributing hosts that carry articles and blog posts, like a newspaper or magazine. This is another feature that makes Medium articles similar to the scientific one. In fact, each "Publication" is made of three kind of contributors: (i) the owner, that is the user that has full rights to manage the "Publication" page, (ii), the editor, that is the one that can review, edit and publish stories submitted by writers, as well as add their own stories and modify the publication's layout; (iii) and the writers, that are regular contributors who can submit their stories to the publication. Once a story is submitted to a publication, the publication editors must then accept the story before it is published into the publication.

These features make the hybrid structure of the Medium platform a good candidate to perform a fast detection of technological convergence for the following reasons:

- the tag system and the publication framework make possible the treatment of articles as scientific publications adopting already known techniques in literature;

- the community involvement, audience engagement and social newsgathering permits a fast availability of original contents.

The choice of an online social network accomplishes previous works showed literature. In fact, the large availability of data makes these platforms a rich source of information, comprising structured, semi-structured and unstructured information, which are vital for various research fields [52,53]. The huge amount of data, mainly textual, that is generated by these platforms can be analyzed at different levels of granularity for various purposes, including behavior analysis, sentiment analysis, and predictive modeling. Furthermore, recent studies demonstrated also how it is possible to observe evolutions in terms of topics discussed over a period of time, for example to track event transitions such as emergence, persistence, convergence, divergence, and extinction [54,55].

In order to clarify how Medium captures the process of technological innovation we compare the trends detected by Medium from other conventional sources, such as patents and publications [56]. We look at the past to see whether and how they correlate. We perform two parallel comparisons.

First, we search for a list of technologies representing the most important ones in the field of Industry 4.0 according to Chiarello [14]. The list includes the following technologies: "autonomous robot", "virtual reality", "internet of things", "cybersecurity", "cloud", "additive manufacturing", "3d printing", "big data", "artificial intelligence", "machine learning". We counted the number of times these technologies appear in (i) the abstract of publications, (ii) the abstract of patents and (iii) the full text of Medium articles. We perform this analysis considering the time frame that starts in 2000 and ends in 2019.

Second, we search for the terms "Industry 4.0" and "Fourth Industrial Revolution" in the abstract of patents and publications and in the full text of the Medium articles. We perform this analysis considering the time frame that starts in 2011 and ends in 2019. This time frame is lagged, given that the two labels were virtually not existing before 2010.

Fig 1 shows the trend lines and the correlation plots for the two analyses. As we can see the trend lines follow a similar path in both cases. In particular, the correlation plots measure the correlation between the trend lines. It is possible to see how the three different sources behave similarly in the context of detecting words that can be considered raw indicators of technological trends.

## 5. A conceptualization of fast technological convergence

Technological Convergence is a complex process, based on the continuous emergence of new ideas but also on the disappearance of old ones, while others give birth to processes of recombination.

The capture of this turbulence has been approached with several methodologies in different fields [57]. Existing studies follow different approaches to track the evolving vocabulary, themes, and topics in online social networks. In this direction Blei et al. [58], presented a state space model-based approach utilizing the natural parameters of multinomial distribution to monitor various topical dynamics over a time-scale. They generated per-document topic distribution and per-topic word distribution for different time-intervals and observed the evolution of a topic over different time-intervals. However, as shown in Di Caro et al. [59], the Dynamic Topic Modeling approach is not able to grasp the birth and death of topics over time and their

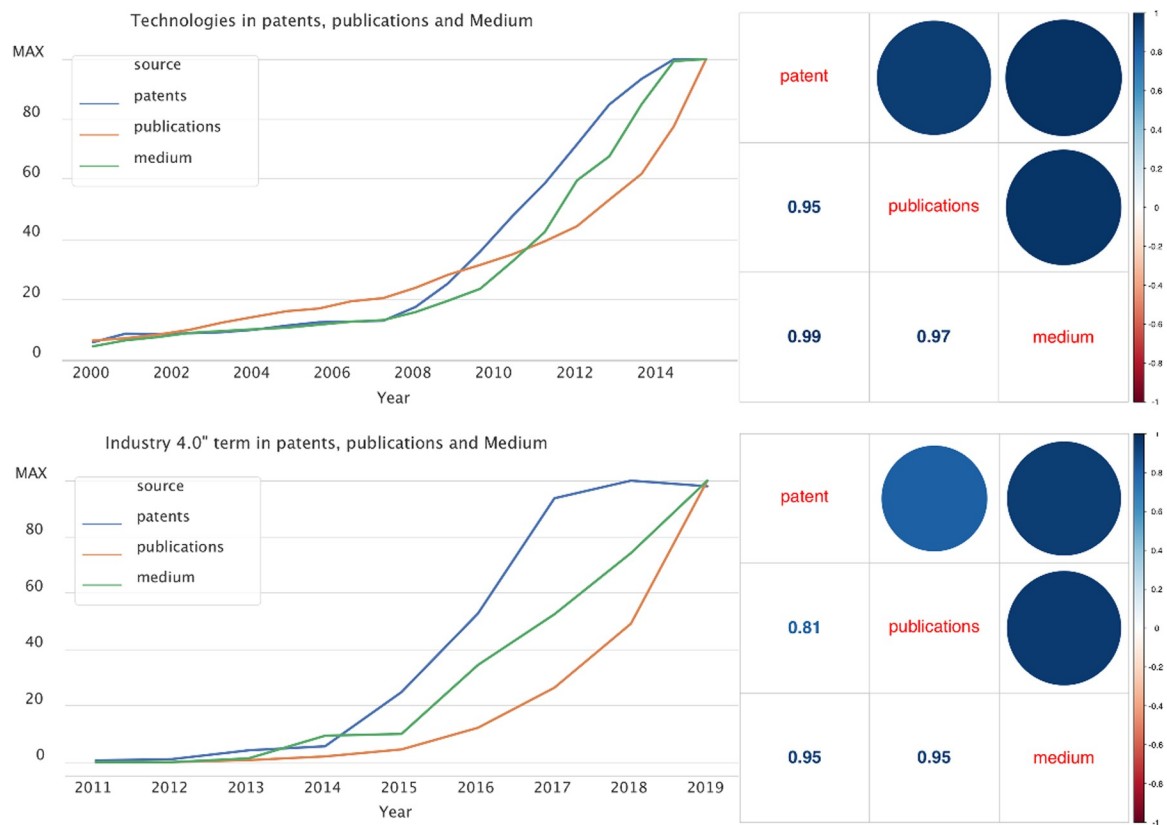

**Fig 1. Comparison of technological trends detected by patents, publications and medium.**

recombination. In particular, they conceptualise knowledge evolution between different time-periods by looking at the transformations occurring between the latent topic structures of the time-windows considered, each of them obtained from running a topic modeling programme.

In this study we worked under the same assumption of Di Caro and co-authors: there exists a structural change of topics between two corpus and it is possible to understand the underlying topic dynamics which explain it. In the context of an online social network, instead of having topic modeling, we compute a network mapping. The possibility of organizing a network under a structure of clusters is similar to a topic model structure.

In this paper we focus on the dynamic evolution of technological clusters over time. Under this assumption, when comparing the clusters generated by a network mapping exercise in two different, although adjacent, time windows, we should be able to capture the evolution of the debate and highlight the birth, death and recombination of clusters. On the one extreme, we can find a situation in which knowledge does not evolve and thus clusters are stable. On the other, we figure out the maximum of turbulence in which new clusters emerge without any semantic relation with the incumbent ones. In the latter case, we may assume the death of past clusters and the birth of new ones. In between the two ideal cases, we can also draw a continuum in which we can observe both deaths and births of clusters. Finally, in a most interesting scenario, rather than observing stability or turbulence, knowledge may evolve recombining existing clusters in both old and new ones. Table 1 summarizes five typical patterns of knowledge evolution and their interpretation.

Fig 2 presents the five ideal types of knowledge evolution as a proximity network of clusters that could be mathematically formalized as follows. Let us consider $M$ clusters emerged as the

**Table 1. Typical pattern of knowledge evolution (source: Di Caro et al. 2017 [59]).**

| | | | |
|---|---|---|---|
| (a) | Stability | a cluster A exists at time t and t+1 |
| (b) | Birth | the cluster A at time t+1 has no antecedent at time t |
| (c) | Death | the cluster A at time t disappears at time t+1 |
| (d) | Merging | multiple clusters at time t combine in a new topic A at time t+1 |
| (e) | Splitting | multiple clusters at time t+1 share an antecedent at time t |

result of a network clustering exercise from a corpus of articles at time *t* and *N* clusters at time *t + 1* from another corpus. We tackle the critical problem of tracking the transformation of the set of clusters *M = (1, ..., A, ..., M)* at *t* into the set of clusters *N = (1, ..., a, ..., N)* at *t + 1*. Specifically, we are interested in measuring the magnitude of the various phenomena such as birth, death, merging, and splitting. In order to do that, consider a similarity index *simil(i,j)* based on word co-occurrence between each couple of clusters *(A, a)* with *A ∈ M* and *a ∈ N* and consider the similarity matrix *S (M × N)*.

$$simil_{i,j} = \frac{|i \cap j|}{|i| + |j|}$$

$$S = \begin{matrix} & & a & \cdots & N \\ A & \begin{bmatrix} simil_{1,1} & \cdots & simil_{1,N} \\ \vdots & & \\ & & \ddots & \\ simil_{M,1} & \cdots & simil_{M,N} \end{bmatrix} \\ M & \end{matrix}$$

We consider the similarity matrix S as the incidence matrix of M over N. We can thus employ S to create a bi-adjacency matrix D, and thus consider Fig 3 as the resulting bipartite network in which M and N are the sets of nodes, while the elements of the matrix are the

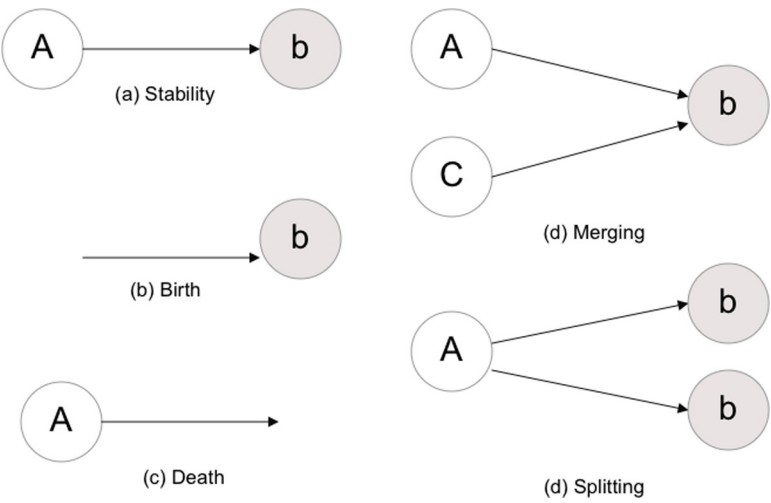

**Fig 2. Typical pattern of knowledge evolution (source: Di Caro et al. 2017 [59]).**

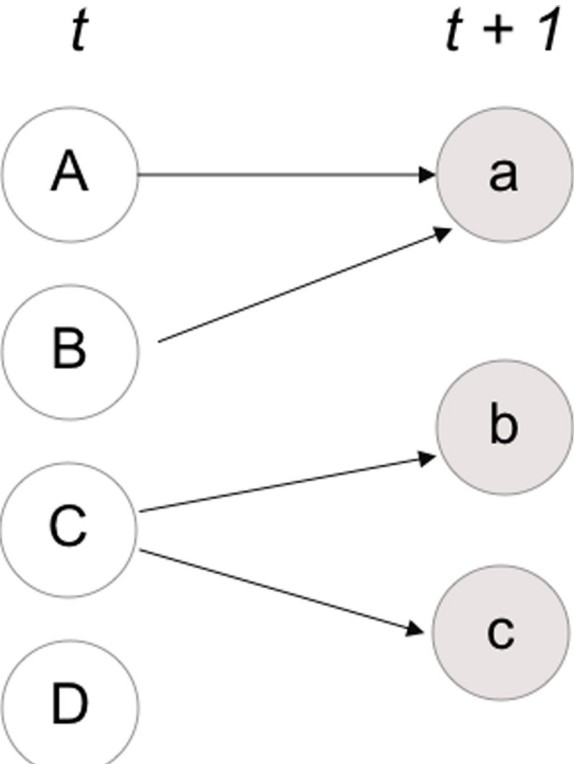

**Fig 3.** Bipartite Network between clusters at time t {A;B;C;D} and clusters at time t+1 {a;b;c}.

weights of the edges.

$$D = \left[ \begin{array}{c|c} 0 & S \\ \hline S^{T} & 0 \end{array} \right]$$

We now show how this representation can help measure the magnitude of births, deaths, merging and splitting. Births and deaths can be easily calculated from the matrix *S*. A row sum equal to zero highlights a death, while a column sum equals zero indicates a birth. A death means that a cluster completely disappears while a birth means that a cluster carries no common nodes with other clusters in the past.

The adjacency matrix allows us to calculate an *Convergence Index (CI)* and a *Novelty Index (NI)*. These two indexes help to understand how much a cluster at time *t+1* inherits elements from the clusters at time *t*.

$$1 = NI_j + CI_j$$

$$NI_j = 1 - \frac{\sum_i^M S_{i,j}}{M}$$

We take the average of all the cell values in matrix S. Since the similarity index is bounded between 0 and 1, *NI* ranges from 0 to 1. For very small values of novelty, new clusters show different nodes from old ones. As mentioned, transformation of clusters can take the form of

merging and splitting. We say that a merging occurs if a cluster at time t + 1 shows a high similarity with two clusters at time *t*, meaning that the nodes of A and B at *t* (as in Fig 3) are present in the cluster a. Similarly, we can say that a split occurs if the nodes of one cluster at *t* are to be found in multiple clusters at *t + 1* as in the case for cluster *C*.

## 6. COVID-19 and fast convergence of technologies

In this section we give evidence of the methodological steps undertaken to detect innovation repurposing due to the COVID-19 pandemic. We can divide the overall methodology in two macro-steps. In the first phase we performed two parallel analysis of Medium articles taking into account two different periods: the time *t*, corresponding to a window before the pandemic, and the time *t+1* corresponding to the period when the COVID-19 occurred. The output of these two parallel analyses are two networks clustered using the Louvain method for community detection. The second, and final step of our method, is the detection and measure of technological convergence according to the conceptualization depicted in Section 5. The media platform used to manipulate and visualize the network and perform the clustering is Gephi [60], an open source and free media platform that performs exploration for all kinds of graphs and networks. The Gephi platform returns graph data in datatables manipulated in python with the *pandas* library. Then, the calculation of the indexes (*CI* and *NI*) has been performed using the python libraries of *numpy* and *scikit-learn*.

### 6.1. Technological clusters before COVID-19

In this step we proceeded with the retrieval of Medium articles concerning a list digital technologies defined by Chiarello et al. [14]. In particular, we used web scraping techniques to download the content and the tags of 43,234 articles produced in Medium of two time windows: January 2018—April 2018 and January 2019—April 2019. We choose these two windows in order to make the comparison between covid-related and non-covid-related articles consistent considering that the spread of the pandemic occurred during the first months of 2020. This triggered a significant convergence of article production towards this theme. This did not occur during the same periods of the previous years. As stated by Rosvall et al. [61], the comparison between changes in large networks requires the analysis between two comparable time windows. So that, we choose to analyze the same time windows for the pre and post COVID.

This document retrieval step is then followed by the extraction phase in which we searched for technologies belonging to a list of technologies defined by Chiarello et al. [14]. This is a list of regular expressions of them, able to detect inflections (e.g. singular, plurals) of the same technologies written in different ways.

After extracting the technologies it was possible to define their link with the other topics addressed within the same articles. This was achieved by calculating the co-occurrence degree (i.e. how many times two elements appear together in the same document) between technologies and tags assigned to each article. The results are then represented as an (N, N) adjacency matrix, where N is the number of unique technologies and the elements in the matrix indicate the number of co-occurrences. We then used the adjacency matrices to generate an undirected graph Gs = (Vs, Es), where the vertices, Vs, are the topics/technologies, and an edge Es, exists if two nodes co-occur at least one time. The weight of each edge is the respective co-occurrence value between the two vertices.

An important issue to take into account for this first step of analysis is the number of nodes. In order to make two networks over two time periods comparable, we need to resize the number of nodes. In practice, the network at time *t* has to have the same number of nodes of the

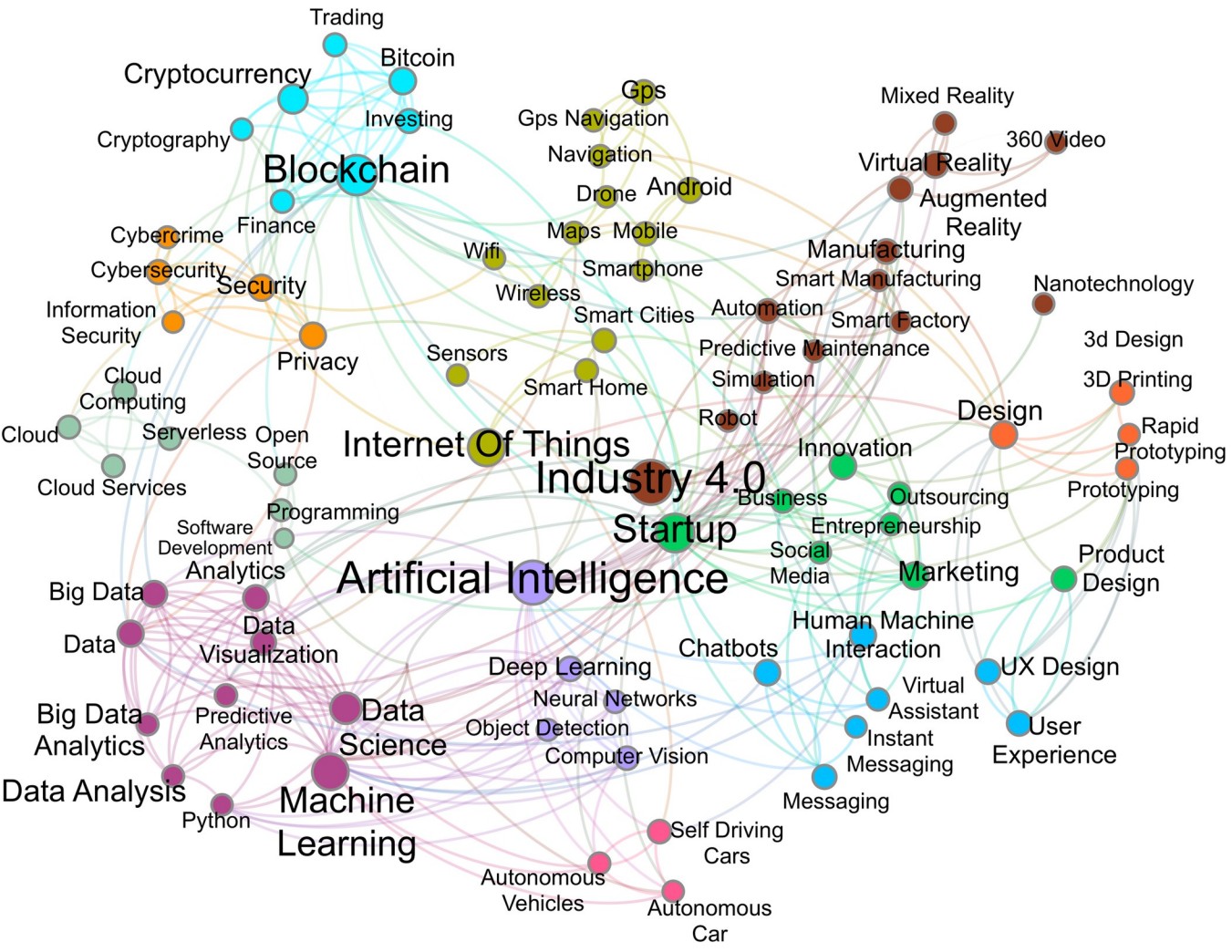

**Fig 4. Pre-covid network graph.**

network at time *t+1*. Under this assumption, we filtered the graph maintaining only the first 100 nodes in terms of frequency of appearance in the articles.

The final step of this phase involves the clustering of the obtained network. The clustering algorithm receives as input the collection of tags and returns a set of terms clusters $C = \{C1, C2, \ldots, Cn\}$ that cover the whole network in analysis. Each cluster $Ci$ is a subset of terms, and a term may belong to only one cluster. Fig 4 visualizes this first graph using the software Gephi with the Force Atlas 2 algorithm [62]. This visualization aims to provide a generic and intuitive way to spatialize networks with a force-directed drawing that has the specificity of placing each node depending on the other nodes. This process depends only on the connections between nodes, in fact two nodes are represented closely if they share an edge. In this way also nodes that belong to the same communities of nodes but do not share any edge are represented closely. In other words, the visualization tends to be coherent with the clustering algorithm. In the figure, the size of each node is proportional to its in-degree while the colour expresses the cluster to which each node belongs. The clustering of this network has been computed using the built-in algorithm of Gephi by calculating the modularity of each node [63]. The process resulted in 11 clusters that have been manually labelled and are showed in Table 2.

**Table 2. Pre-covid clusters.**

| Cluster name | Top 5 tags |
|---|---|
| Business | business, startup, social media, marketing, innovation |
| Blockchain | blockchain, bitcoin, cryptocurrency, trading, finance |
| Data Science | data science, machine learning, data analytics, data, big data |
| Manufacturing | industry 4.0, smart manufacturing, automation, virtual reality, augmented reality |
| Deep Learning | deep learning, artificial intelligence, neural networks, computer vision, object detection |
| Internet of Things | internet of things, wifi, sensors, android, mobile |
| Human Machine Interaction | human machine interface, chatbot, UX design, user experience |
| Cybersecurity | cybersecurity, security, privacy, information security, cybercrime |
| Cloud | cloud, cloud computing, cloud services, programming, open source |
| 3D Printing | 3d printing, rapid prototyping, design, 3d design, prototyping |
| Autonomous Cars | autonomous car, self-driving car, autonomous vehicle, transportation, automotive |

## 6.2. Technological clusters after COVID-19

After building a network that depicts the technological scenario before the pandemic, we use the same approach described in section 5.1 to build a network from Medium articles but this

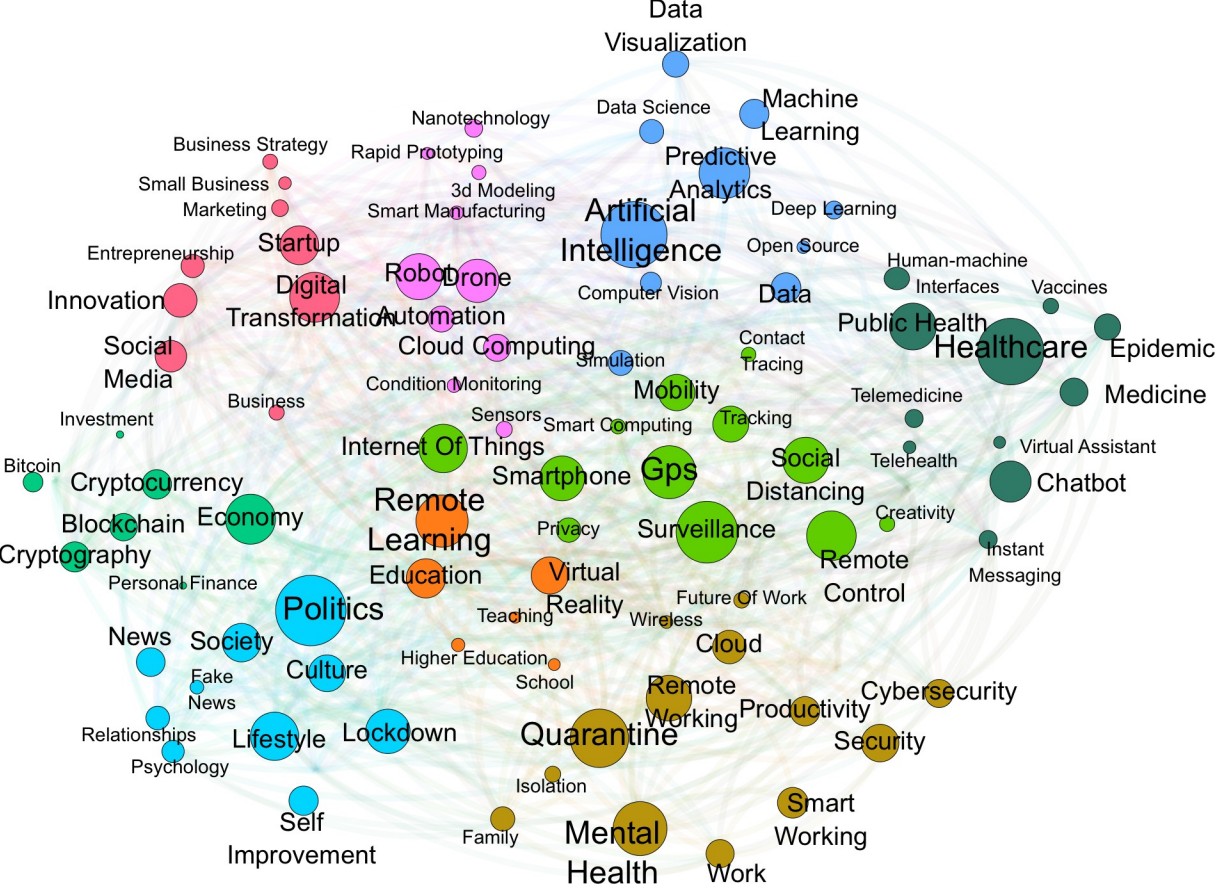

**Fig 5. Post-covid network graph.**

**Table 3. Post-covid clusters.**

| cluster name | Top 5 tags |
|---|---|
| Digital Marketing | digital transformation, digital marketing, social media, startup, innovation |
| Fintech | economy, cryptocurrency, cryptography, blockchain, bitcoin |
| Artificial intelligence | artificial intelligence, predictive analytics, machine learning, data visualization, deep learning |
| Automation | automation, robot, drone, cloud computing, condition monitoring |
| Remote Learning | remote learning, education, virtual reality, higher education, school |
| Remote Control | internet of things, remote control, surveillance, social distancing, gps, smartphone |
| Healthcare | healthcare, chatbot, public health, virtual assistant, medicine |
| Social | politics, lockdown, self improvement, lifestyle, psychology |

time during the spread of COVID-19. We proceeded with the retrieval of all documents inherent to the pandemic. More specifically, we searched for articles that had the "covid-19" tag and subsequently, used web scraping techniques to download the related content, i.e. title of the articles, body and tags. This process was mainly facilitated by the structure of Medium's web pages.

We then filtered those articles that mention a technology. As for the pre-covid network, we used the list of technologies [14] to detect what are the articles that talk about the pandemic and mention a technology.

The building and subsequent clustering of the post-covid network follows the same assumptions and steps of the previous one. Fig 5 shows the network whose node colour expresses the cluster to which the node belongs, while Table 3 shows the resulting 9 clusters that are manually labelled.

## 6.3. Technological convergence

Our conceptualization of fast technological convergence (see Section 4) brings us to show the results we obtained in our experiment. First, we build the Similarity Matrix $S$ between the clusters of the two networks (see Fig 6).

The Similarity Matrix shows on the rows the clusters of the pre-covid network and on the columns the clusters of the post-covid network. The next step is the building of the bi-adjacency matrix that allows to represent the bipartite network of clusters of two time windows (see Fig 7).

This bipartite network captures the phenomena that we assumed in our conceptualization of technological convergence. In particular, the colours of the nodes express the scale of the phenomenon. For example, the "Blockchain" cluster in the pre-covid network only has nodes in common with the "Fintech" cluster of the post-covid network. This means that the technology exists in both time windows. The same for the "Business" and "Human Machine Interaction". For what concerns the "Autonomous Car" and "Social" clusters, we have witnessed the two phenomena of death and birth. This is because the two clusters have no common node respectively in the two time windows.

However, Fig 7 does not capture adequately the relations between the two time windows. In particular, the number of nodes that two clusters share belongs only to a small part of the overall nodes of the cluster at time $t+1$. This could be better explained by the *Novelty Index* (NI) and the *Convergence Index* (CI).

In order to better understand this affirmation, let us consider the example of the cluster "Remote Control". Fig 7 shows evidence of merging for this cluster. This is because it appears after the pandemic and has the similarity values of 0.32 and 0.15 respectively with the pre-

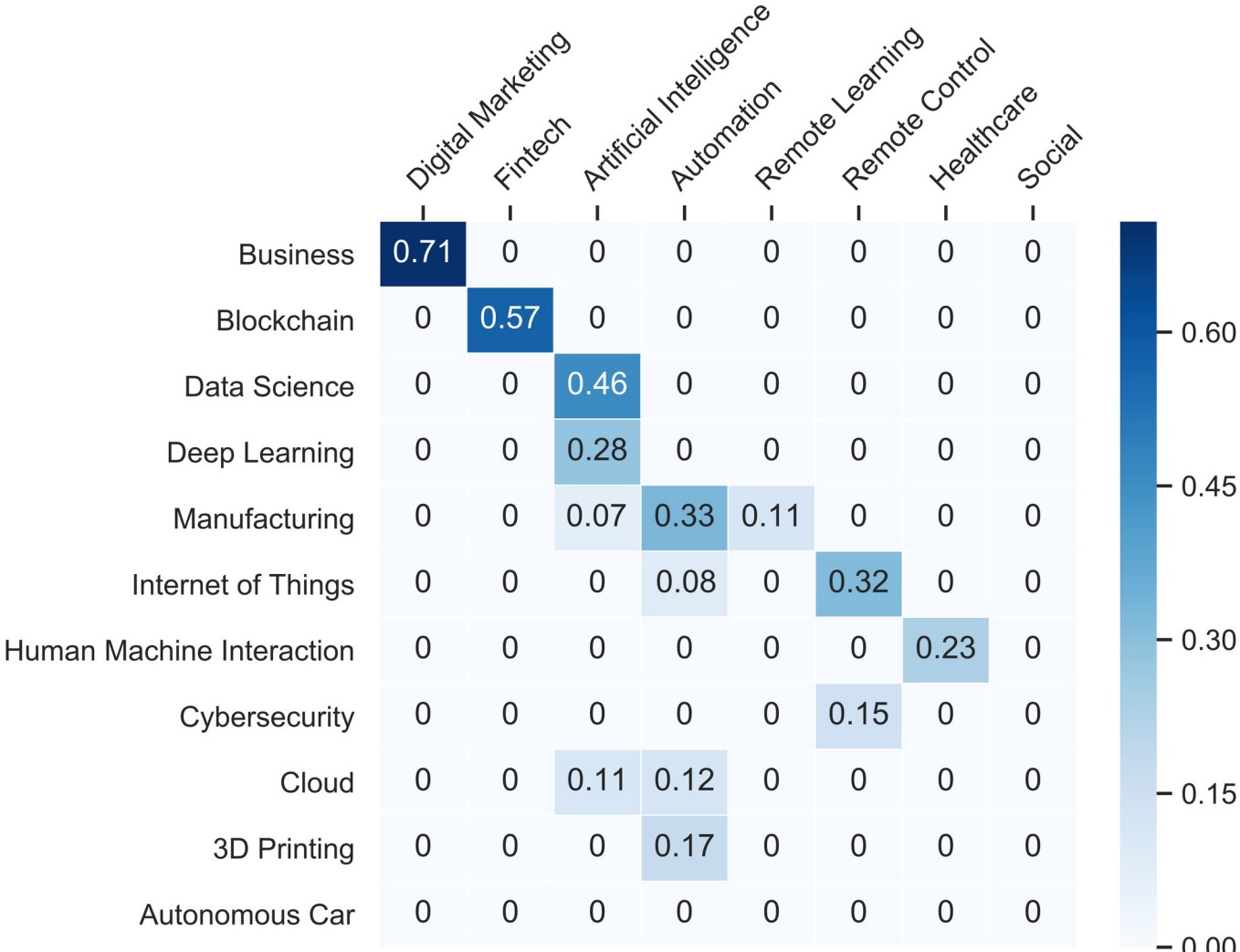

**Fig 6. Similarity matrix between clusters of pre and post covid.**

covid clusters "Internet of Things" and "Cybersecurity". These two clusters do not merge with any other cluster of post-covid.

However, saying that "Remote Control" is the intersection of "Internet of Things" and "Cybersecurity" is not totally correct. In fact, only a part of the "Remote Control" nodes are present in the pre-covid network. In fact, the *Convergence Index* of this cluster is 0.46. This means that 46% of the "Remote Control" nodes are present in the pre-covid network, while the remaining 54% are totally new. Therefore, one interpretation we can give is that the "Remote Control" cluster is half coming from the "Internet of Things" and "Cybersecurity" and the other half is totally new. Fig 8 shows the values of the CI and NI for each cluster of the post-COVID network.

Once we have the clusters for the networks at each time, we want to reveal the trends in our data: we need to simplify and highlight the structural changes between clusters [63]. In the mapping-change step of Fig 9, we show an alluvial diagram that highlights and summarizes the structural differences between the pre and post covid clusters. Each cluster in the network is represented by an equivalently colored block in the alluvial diagram. Changes in the clustering

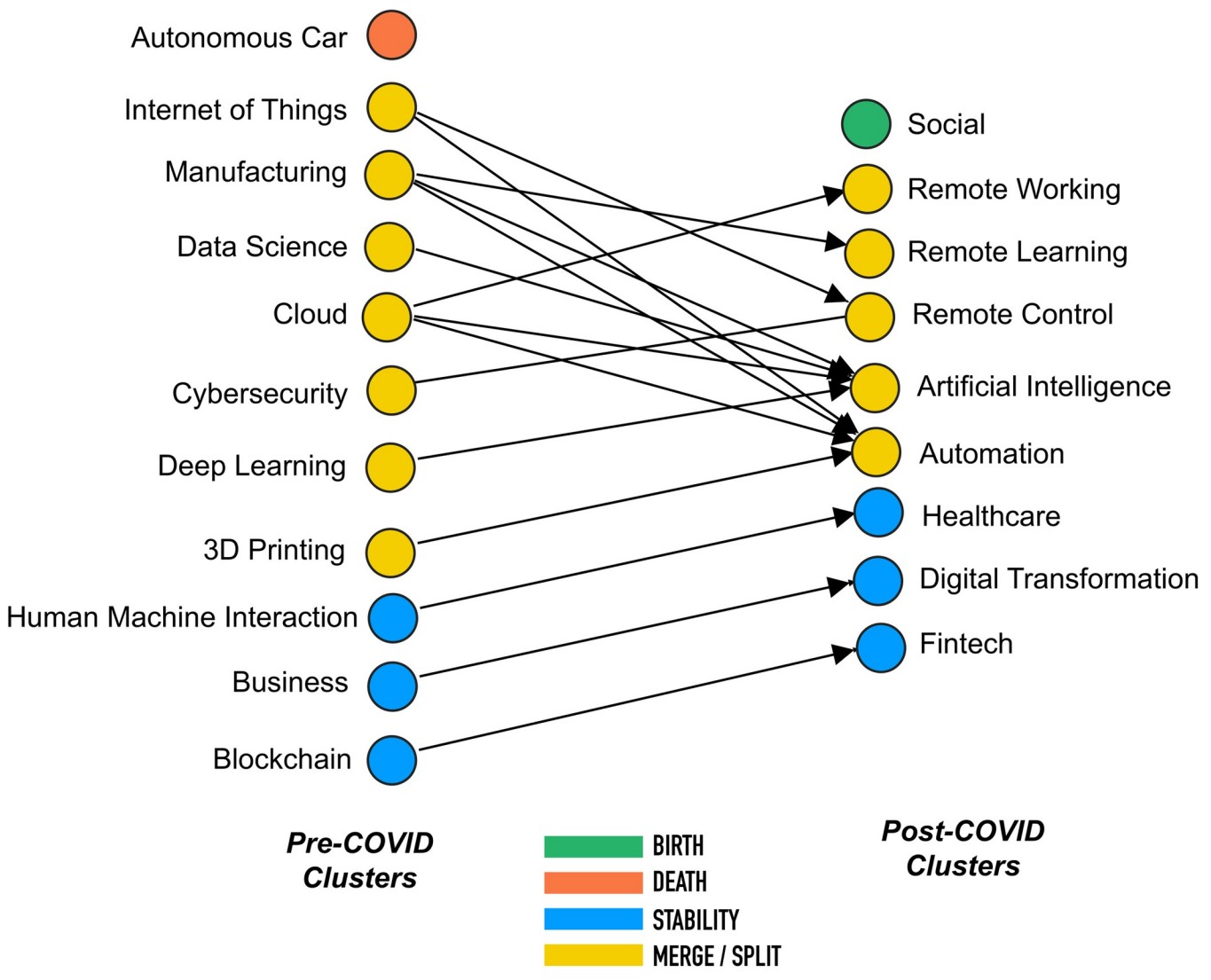

**Fig 7. Bipartite network of pre and post covid clusters.**

structure from one time period to the next are represented by the mergers and divergences that occur in the ribbons linking the blocks at time *t* and time *t+1*.

In Fig 9 it is possible to observe what are the most important elements that relate the pre-covid network to the post-covid one. For example, the post-covid cluster of "Healthcare" is related to the pre-covid cluster of "Human Machine Interaction" because it shares the nodes of "Instant Messaging" and "Virtual Assistance". However, with this figure we want to obviate a critical aspect of our methodology, that is the cluster labeling. Cluster labeling is the problem of picking descriptive, human-readable labels for the clusters produced by a clustering algorithm. In the case of this experimental setup, this process is very critical for the clarity of the reader. In fact, it is not enough to describe the cluster by their label to explain such convergence phenomenon. Furthermore, the clustering algorithms used in this experimental study do not produce any such labels. To tackle these problems we involved a panel of experts made of one professor of Mechanical Engineering, a postdoc in Engineering Design and two PhD

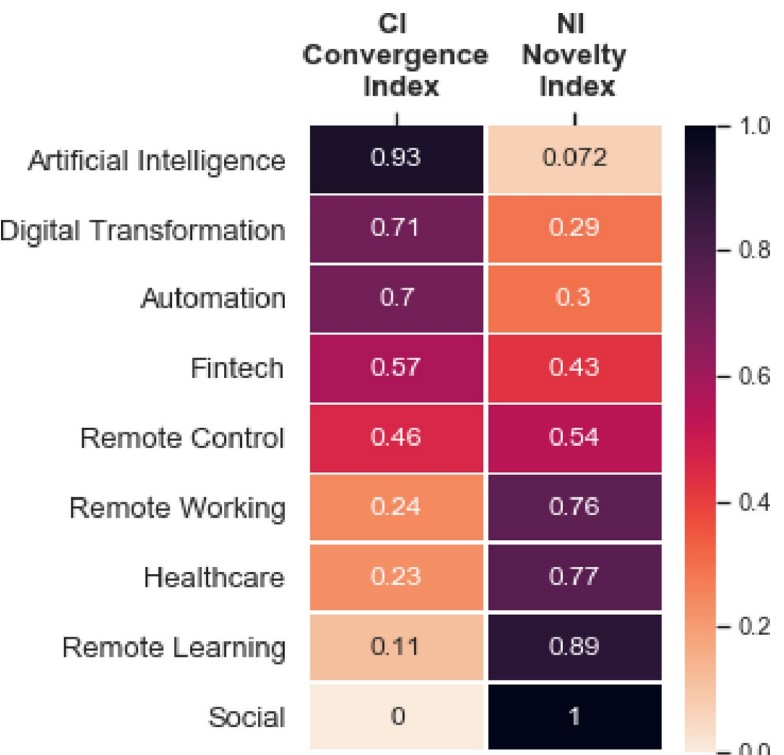

**Fig 8. Convergence Index (CI) and Novelty Index (NI) for post-covid clusters.**

students in "Smart Industry" that examine the elements per cluster to find a labeling that summarize the theme of each cluster and distinguish the clusters from each other.

The results of this experimental study show a remarkable phenomenon of rapid technological convergence that occurred in the first quarter of 2020. We dealt with two metrics, *CI* and *NI*, to measure this phenomenon. The final step for this experimental study is to argue that the coronavirus pandemic impacted drastically on these two metrics. Hence, we need to demonstrate that the fast technological convergence occurring in this restricted time frame has been stronger than before. We observe the values of the two metrics over a period of time (from January 2017 to April 2020, namely 10 quarters) building the time series shown in Fig 10. The behaviour of these measures shown in the time series suggests a clear discontinuity occurring between the last two quarters of September 2019—December 2019 and January 2020—April 2020. In order to confirm the impact of the COVID-19 on these measures we argue the following considerations.

First, both the *CI* and the *NI* do not depend on the number of articles produced. The formulas used for their calculation (see Section 5) do not take into account the production rate in terms of number of articles. This leads to state that our analysis does not depend on scale factors. For example, if a growing interest on the platform occurred, a greater number of articles would have been produced. Another example of increased production could be given by the introduction of a new technology that would generate more interest and therefore the involvement of a greater number of authors who would consequently produce more articles. All these scenarios would not have impacted on the measure of *MI* and *NI*.

Consequently, the sudden increase in a relative short time frame of the indicators admits the influence of other factors of world scale, such as the coronavirus pandemic. Other plausible explanations, such as endogenous technological developments or increase in research funding, must be ruled out, as they typically have a much longer time window. The abrupt change occurred in the

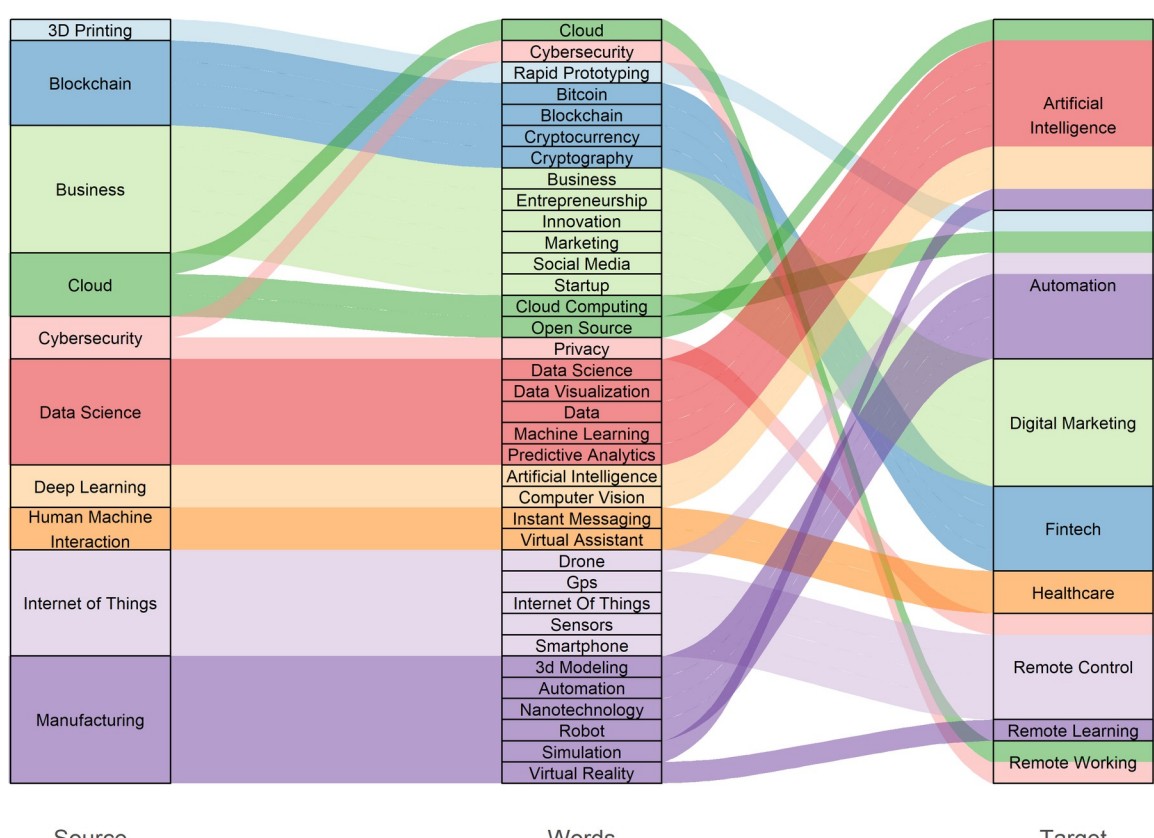

**Fig 9. The alluvial diagram, with clusters ordered by size, reveals changes in network structures over time. Here the height of each block represents the volume of flow through the cluster.**

last observation of time series (Fig 10) suggests the manifestation of a structural break [64]. In econometrics and statistics, a structural break is an unexpected change over time in the parameters of regression models built upon the observation of a time series. We are in the case of a single break in mean with a known breakpoint. For this reason we can employ the Chow test [65]. This test is built on the premise that if the parameters of a time series are constant then out-of-sample forecasts should be unbiased. In this case, we test the null hypothesis that there is no structural break in the observation between the last two quarters. The test procedure first estimates coefficients for each period and then uses the out-of-sample forecast errors to compute an F-test comparing the stability of the estimated coefficients across the two periods. We compute the Chow Test using the python package *chowtest* (package available at the following link: https://pypi.org/project/chowtest). For both the *Novelty Index* (Chow Statistic = 131.8; p-value = 0) and *Convergence Index* (Chow Statistic = 98.8; p-value = 0) we reject the null hypothesis of equality of regression coefficients in the last two periods. This test leads to the conclusion that a structural break occurred between the end of 2019 and the beginning of 2020. Following the considerations discussed above we can conclude that a plausible explanation for this structural change is given by the effect of the coronavirus pandemic on the fast recombination of technologies.

## 7. Discussion

What we observe is a striking phenomenon. By filtering the papers devoted to Covid-19 with a list of digital technologies [14] we obtain a map (Fig 9) in which it is evident the presence of a

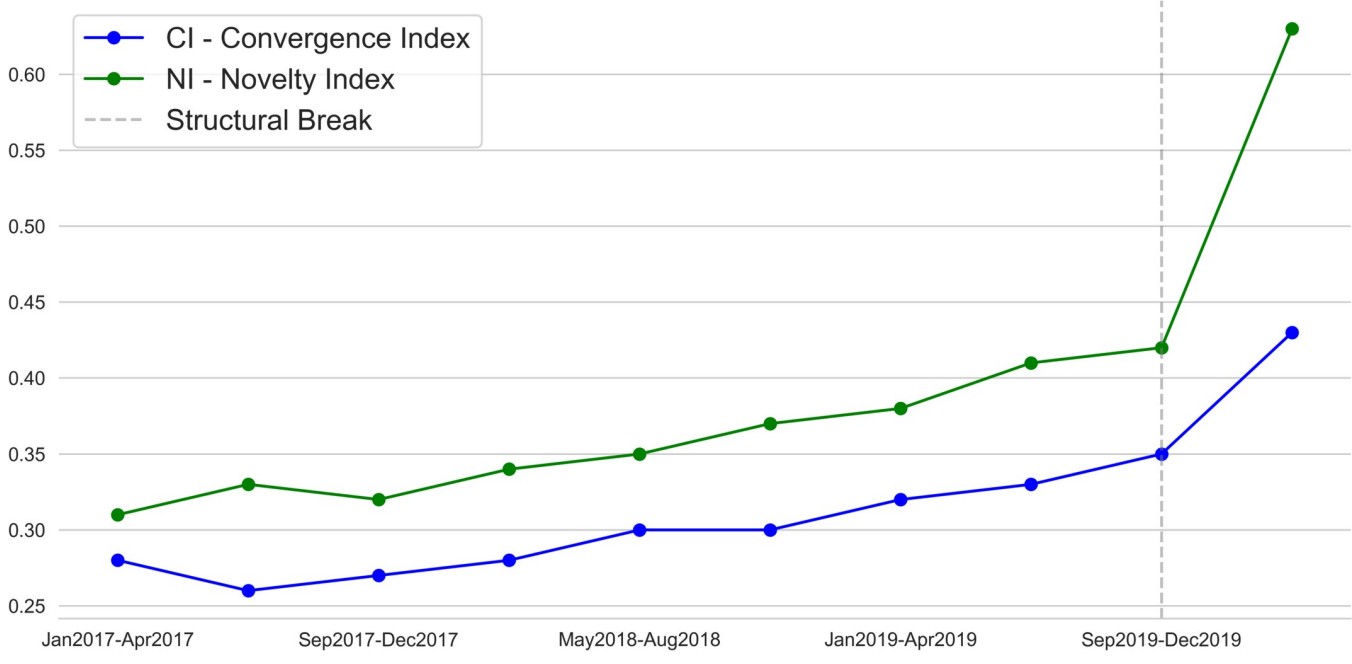

**Fig 10. Time series of Convergence Index (CI) and Novelty Index (NI).**

central core of technologies and several peripheral nodes. The core, as expected, is formed by all technologies that are in the perimeter of Industry 4.0.

However, by clustering the co-occurrence network of keywords we identify several non-core regions that refer to new areas of application of the same technologies in the Industry 4.0 core. We interpret these regions as evidence of the fast convergence of the technological community towards applications that share the core technologies of cyber-physical systems, while developing specific solutions for the social problems created by the pandemic.

Among the top eight clusters after Covid we identify the newly created clusters:

- Remote control

- Remote working

- Health

- Remote learning

We first examine in detail these areas. In all these areas the main technologies were already available before the Covid crisis but the pandemic bringed new extreme requirements and thus new solutions. Moreover, these technologies have been developed in different sectors: the medical device industry for telemedicine applications, specialized software industries for E-Health, smart working and distance education applications, the ERP and database industries for e-commerce applications, the finance and insurance industries for Fintech.

## 7.1 Remote control

Remote control includes all technologies for the operation of machines and equipment at a distance, under strict conditions of controllability and safety. Remote activities include maintenance, testing, and operation. For example, Fabio Perini, a leading company in paper and tissue equipment manufacturing, has developed a system for remotely controlling the activities of 300 client

sites worldwide, offering distance support for maintenance and repair [66]. The acquisition of manufacturing automation firms since 2017, as recently discovered by the specialized press [67], may explain the somewhat controversial decision of Tesla to keep the plants open in the US.

Within the Industry 4.0 framework it was mainly intended as an auxiliary technology, not completely substituting but complementing the traditional industrial control of operations. With the lockdown of workers and the subsequent establishment of strict safety measures for physical distancing of workers it has become a crucial technology for assuring business continuity. In the post-Covid landscape this technology gained prominence.

The Remote control cluster benefits from inputs of IoT and Cybersecurity. IoT establishes the cyber-physical connection between manufacturing operations and data (digital twins). On the other hand, industrial data are sensitive and proprietary and cannot be processed unless a comprehensive framework for cybersecurity is established. Under Remote control, in fact, a distributed network of home-based computers must be habilitated to interact with central manufacturing operations.

Although the two technologies are not the same, Remote control is facilitated by the level of robotisation of plants. There is evidence that the use of robots decreases the risk of infection for workers. Yang et al. [68] offer several examples of use of robots to combat contamination. On the basis of a classification of industrial sectors at 4 digits for the Italian industry and data on the intensity of robot installations, Caselli et al. [69] show that the exposure to contagion risk is negatively related to robotisation.

## 7.2 Remote working

The lockdown created a sharp divide between economic activities that should be kept open (essential sectors) and activities that have been shut down. Companies have managed the lockdown by shifting work activities at home (Work from home, WFH).

Work from home can be done for most service activities and for the administrative functions of manufacturing activities. Boeri et al. [70] have proposed a classification of home workers in three types: Type 1 (complete WFH), Type 2 (WFH and limited mobility but no face-to-face interaction) and Type 3 (WFH, mobility and limited interaction). Dingel and Neiman [71] and Mongey et al. [72] have used the US O*Net survey to classify work activities in terms of their potential for WFH. Overall they estimate that complete WFH can be done for 30–35% of work positions, while WFH with limited mobility may achieve approximately 50% of workers. On the basis of German work survey data Fadinger and Schymik [73] reach similar estimates. The potential for WFH is, however, drastically reduced in developing countries, due to poor availability of domestic devices (PC, tablet) and lack of broadband connectivity. Saltiel [74] and Delaporte and Peña [75] argue that the actual potential in these countries is in the order of 10–20% maximum of work activities and Hatayama et al. [76] show that the amenability to working from home is positively related to economic development. All these studies agree that WFH will be a permanent feature of the organization of work in the near future.

From a technological point of view, WFH requires a well established set of solutions: PC or tablet, broadband Internet connection, webcam, software suites for video call and groupwork. All these technologies were available before the crisis. However, several improvements have been rapidly proposed and introduced. Among them: Software solutions to optimize the quality of videocall and meetings (e.g. by focusing the screen on those who take the word in a meeting); Software tools to manage distributed teamwork (e.g. solutions to work in parallel to documents); Cyber-security tools to ensure security in a distributed and remote work environment; Tools to control the remote work; Artificial intelligence and chatbot applications for the facilitation of the work of call centers [77] and the reduction of work stress [78].

These technological developments will be crucial in the future work landscape. It in clear, in fact, that after having experienced the benefits of WFH, most workers will ask to continue even after the Covid crisis [79,80]. The future landscape will be one of hybrid work organization, partly in presence, partly at distance.

## 7.3 Health

While most of the attention of researchers has been directed toward the discovery of a vaccine against the coronavirus and the testing of existing drugs, a smaller community of medical technologists has been active in the development of solutions for distance medicine. Telemedicine has a long technological history, but has traditionally suffered from lack of implementation. According to Oudshoorn [81] telemedicine has been introduced as a cost cutting care approach, without a proper consideration of many patients needs such as the need for personal, face-to-face contacts with healthcare providers. The design of telemedicine solutions has been done ignoring socio-technical requirements analysis [82] and a comprehensive understanding of the delivery process [83,84]. Only recently the care delivery service has been designed on the basis of use cases of more complex care needs [85,86] (see [87] and [88] Luzi for the cases of asthma and long term ventilation).

This has limited the social acceptance of advanced health care technologies, such as Personal Health Systems (PHS) [89,90], as well as the Electronic Health Record (EHR) systems [91,92].

In the last few years there have been impressive advancements in technologies aimed at reducing the distance between face-to-face interaction and screen-mediated presence. At the same time, the lockdown has forced home-based patients suffering from chronic diseases to arrange frequent diagnoses at a distance. The large gulf between distance interaction with healthcare providers and personal contact has been greatly reduced thanks to the convergence between the pressure of urgent social needs (providing care under the lockdown) and the opportunities opened by technological advancements.

On the other hand, there are areas in which technological developments have not been exploited appropriately, given the lack of an institutional environment. The need for advanced e-health systems has been made explicit by the difficulty to share patient-level records on the virus during the pandemia. Hospitals have joined their forces in a voluntary way by sharing laboratory data, but in the absence of an internationally validated data platform infrastructure. The lesson from this unbalance is important. Health care is both a medical and a social technology [93].

The health care sector has a big potential for applications of Industry 4.0 technologies. These are oriented towards patient-centered remote operations in order to deliver a care delivery experience which resembles the personal interaction with health care providers. On the basis of health care delivery at a distance, patients might be willing to share their clinical data within a fully secure environment, feeding large systems of E-Health record management and clinical data sharing at international basis.

Finally, on top of E-Health data management systems there would be huge opportunities to apply Artificial Intelligence for predictive medicine applications. Von Krogh [6] documents the use of computational biology and Machine learning technologies for the ultrafast repurposing of existing drugs.

The cyber-physical integration between human care and IT can be considered what Industry 4.0 is about, mutatis mutandis. The rapid convergence of the health care industry towards Industry 4.0 applications during the Covid crisis is most likely a predictor of future large scale implementations.

## 7.4 Remote learning

One of the most challenging social implications of the lockdown has been the sudden and complete shutdown of schools and universities. This has created a huge demand for remote education. The most immediate reaction has been the shifting of traditional lectures and teaching materials from the classroom to the screen, using one of the platforms made available by large IT companies.

Over the weeks, however, it has become more and more evident that this approach to education is not simply a change in delivery modes, but involves a complete reconfiguration of the education process [94]. The degree of attention of scholars rapidly deteriorates. Teachers miss the feedback from the class, in which non-verbal clues are fundamental. The students that show low engagement in school activities, which in the traditional context of the classroom are the special attention of teachers, are now hidden behind the screen and their detachment may become undetected and uncorrected. This will require, for the months and years to come, heavy investment into new educational technologies and skills.

At the same time, improvements in remote learning technologies that reduce the gap between face-to-face and remote experience will create a serious threat to educational institutions, in particular universities [95].

## 8. Conclusion

In this paper we proposed a novel methodology to perform a fast detection of fast technological convergence. The scenario brought by the covid-19 crisis has demonstrated the need for a rapid technological repurposing and the need for a fast detection of this phenomenon. We tackled these two problems and we give a twofold contribution to the literature.

First, the fast detection has been performed thanks to the use of a novel source: the online blogging platform of Medium. It has been demonstrated that it is suitable for our purpose for two reasons: first, it presents a structure with tags and publications that is similar to academic scientific production; second, it is largely adopted as a platform for social journalism, that is fast and open.

Second, the fast technological convergence has been conceptualized using a network-based technique. We defined a quantitative method that captures this phenomenon by analysing the differences of the networks between two time periods, that in this case were the one before and the one after the pandemic.

The results of these experiments gave us the possibility to discuss the fast technological convergence occurred in the period of spread of covid-19. In particular, we have been able to detect and discuss the repurposing of technologies regarding "Remote Control", "Remote Working", "Health" and "Remote Learning".

However, this methodology has limitations in its two contributions. For what concerns the fast detection, the substantial difference between Medium and other typical sources (such as publications, patents etc.) is the reliability of the content. If academic articles are peer-reviewed by other academics that are experts in the field, in the Medium articles the review is performed by other editors that could not have been recognized as experts in the scientific community. For what concerns the conceptualization of fast technological convergence, the assumption that two networks of the time frame has to have the same number of nodes can be trivial. It has been demonstrated [61] that if we want to describe a phenomenon using networks we cannot limit the number of nodes, especially if we want to perform statistical measures, such as clustering. In this methodology, we choose to resize the two networks with an arbitrary number of

nodes (100) in order to make them comparable. This resizing task may have hidden some important network nodes for an in-depth analysis.

On the other hand, this methodology brought several advantages in terms of rapidity of detection. Unlike traditional sources (such as publications and patents) we have been able to capture the manifestation of technological repurposing analysing a source with low latency of publication. Furthermore, the conceptualization of fast technological convergence does not depend on the source used. Our assumption of similarity between Medium articles and scientific manuscripts, in particular, the presence of the tags or keywords makes this conceptualization suitable for a further application to publications or patents.

## Supporting information

**S1 Raw data.**
(ZIP)

## Author Contributions

**Conceptualization:** Nicola Melluso, Andrea Bonaccorsi.

**Data curation:** Nicola Melluso.

**Formal analysis:** Nicola Melluso.

**Funding acquisition:** Andrea Bonaccorsi.

**Investigation:** Andrea Bonaccorsi.

**Methodology:** Nicola Melluso.

**Supervision:** Andrea Bonaccorsi, Gualtiero Fantoni.

**Validation:** Filippo Chiarello.

**Visualization:** Filippo Chiarello.

**Writing – original draft:** Nicola Melluso, Andrea Bonaccorsi.

**Writing – review & editing:** Filippo Chiarello, Gualtiero Fantoni.

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
