## [Decision Letter · Decision Letter 0]

9 Sep 2020

PONE-D-20-21013

Rapid detection of fast innovation under the pressure of COVID-19

PLOS ONE

Dear Dr. Bonaccorsi,

Thank you for submitting your manuscript to PLOS ONE. After careful consideration, we feel that it has merit but does not fully meet PLOS ONE’s publication criteria as it currently stands. Therefore, we invite you to submit a revised version of the manuscript that addresses the points raised during the review process.

The manuscript requires several revisions towards literature review and quantitative framework. Furthermore, there are concepts such as "resistance" or "resilience" that should be explained.

We look forward to receiving your revised manuscript.

Kind regards,

Stefan Cristian Gherghina, PhD. Habil.

Academic Editor

PLOS ONE

Journal Requirements:

2.Thank you for stating the following financial disclosure:

 [NO].

Reviewers' comments:

Reviewer's Responses to Questions

**Comments to the Author**

1. Is the manuscript technically sound, and do the data support the conclusions?

Reviewer #1: Yes

Reviewer #2: Partly

2. Has the statistical analysis been performed appropriately and rigorously? 

Reviewer #1: Yes

Reviewer #2: No

3. Have the authors made all data underlying the findings in their manuscript fully available?

Reviewer #1: Yes

Reviewer #2: Yes

4. Is the manuscript presented in an intelligible fashion and written in standard English?

Reviewer #1: Yes

Reviewer #2: Yes

5. Review Comments to the Author

Reviewer #1: This manuscript addresses the issue of technological convergence as a consequence of the COVID-19 crisis. In order to fulfill that aim, the authors perform a network analysis of reseach clusters based on the website Medium. Their findings suggest that the health crisis has fostered a process of convergence in technological innovation. Overall, the paper means an interesting piece of research I consider publishable in a journal like PLOS ONE. The article is informative and useful, concise and well focused and it mostly reads very well. The methodology comprises a remarkable degree of originality, combining the use of Medium and the network analysis. The structure is very appropiate and the conclusions are well supported by the empirical findings. I would like to highlight the analysis of the pre-COVID-19 period, when they show that the process of convergence observed later was not present earlier. Therefore, they are able to claim some causal association between the health crisis and the process of convergence observed later. In any case, after reading the manuscript carefully, I have some comments that, hopefullly, will help to improve the quality of the manuscript.

Firstly, the authors themselves recognizes that the use of Medium implies some advantages and limitations. Although it allows overcoming the shortcomings, in terms of time, due to patents and publications, it is not clear how Medium captures the process of technological innovation. If it is not very costly, I would suggest to try to compare the trends detected by Medium and from other conventional sources (patents or publications) in the past to see how they correlate and whether, in this way, we can reinforce the convenience of using this social network. It is possible that some research work has done so, in that case, it would be enough to cite it.

Secondly, from my point of view, the article would benefit from reinforcing the reasons for which measuring technological convergence is so relevant for society. The authors mentions this issue in the introduction, but I think that it is not enough clear and it is partly taken for granted. For readers not so familiarised with the literature, it would be good to state more clearly some reasons why measuring technological convergence is important, providing some examples.

Lastly, there are some concepts in the introductory section (like "resistance" or "resilience") whose meaning is not very precise. I would suggest reformulating them.

Reviewer #2: Report for “Rapid detection of fast innovation under the pressure of COVID-19”

General comments:

This paper presents a novel methodology to perform a rapid detection of the fast technological convergence phenomenon that is occurring under the pressure of the Covid-19 pandemic. The technological convergence phenomenon has been modelled through a network-based approach, analysing the differences of networks computed during two time periods (pre and post COVID-19). The results led us to discuss the repurposing of technologies regarding “Remote Control”, “Remote Working”, “Health” and “Remote Learning”. The research methods are basically reasonable. Specifically, there are several suggestions to help the author improve the quality of the paper：

1. It is suggested that in the part of introduction or research contribution, if the relevant literature of covid19 reflects the marginal contribution of this study, the following papers on covid19 can be cited.

Shen H, Fu M, Pan H, et al. The Impact of the COVID-19 Pandemic on Firm Performance[J]. Emerging Markets Finance and Trade, 2020, 56(10): 2213-2230.

Fu, M., & Shen, H. (2020). COVID-19 and corporate performance in the energy industry. Energy RESEARCH LETTERS, 1(1): 12967. https://doi.org/10.46557/001c.12967

Qin X, Huang G, Shen H, et al. COVID-19 Pandemic and Firm-level Cash Holding—Moderating Effect of Goodwill and Goodwill Impairment[J]. Emerging Markets Finance and Trade, 2020, 56(10): 2243-2258.

2. The lack of literature review on covid19 leads to inadequate theoretical research contribution. The above literature on covid19 can be cited.

3. This paper does not introduce which media platforms the technology clustering comes from

4. What network platform data are used to build the novelty index (Ni) and the convergence index (CI)?

5. This paper only compares the first quarter of 2019 with the first quarter of 2020. Why not the first quarter of 2020 and the fourth quarter of 2019? Or is it not a comparison of 2016, 2017, 2018, 2019 and 2020?

6. After obtaining Ni and Ci, can we use the difference in difference (did) method in Shen h, Fu m, pan h, et al. The impact of the coved-19 pandemicon firm performance [J]. Emerging Markets Finance and trade, 2020, 56 (10): 2213-2230? Otherwise, without controlling other influencing factors, how can the author judge that these influences must be brought by covid19 instead of other factors?

6. PLOS authors have the option to publish the peer review history of their article (what does this mean?). If published, this will include your full peer review and any attached files.

Reviewer #1: **Yes: **José-Ignacio Antón

Reviewer #2: **Yes: **huayu shen

---

## [Author Response · Author response to Decision Letter 0]

3 Nov 2020

Reviewer #1: 

This manuscript addresses the issue of technological convergence as a consequence of the COVID-19 crisis. In order to fulfill that aim, the authors perform a network analysis of reseach clusters based on the website Medium. Their findings suggest that the health crisis has fostered a process of convergence in technological innovation. Overall, the paper means an interesting piece of research I consider publishable in a journal like PLOS ONE. The article is informative and useful, concise and well focused and it mostly reads very well. The methodology comprises a remarkable degree of originality, combining the use of Medium and the network analysis. The structure is very appropiate and the conclusions are well supported by the empirical findings. I would like to highlight the analysis of the pre-COVID-19 period, when they show that the process of convergence observed later was not present earlier. Therefore, they are able to claim some causal association between the health crisis and the process of convergence observed later. In any case, after reading the manuscript carefully, I have some comments that, hopefullly, will help to improve the quality of the manuscript.

Firstly, the authors themselves recognizes that the use of Medium implies some advantages and limitations. Although it allows overcoming the shortcomings, in terms of time, due to patents and publications, it is not clear how Medium captures the process of technological innovation. If it is not very costly, I would suggest to try to compare the trends detected by Medium and from other conventional sources (patents or publications) in the past to see how they correlate and whether, in this way, we can reinforce the convenience of using this social network. It is possible that some research work has done so, in that case, it would be enough to cite it.

We thank the reviewer for the comment. We performed two comparative analysis of the technological trends detected in patents, publications and Medium. We discussed this analysis in detail in Section 4 showing the trends and correlations that confirm that Medium is able to detect technology trends.

Secondly, from my point of view, the article would benefit from reinforcing the reasons for which measuring technological convergence is so relevant for society. The authors mentions this issue in the introduction, but I think that it is not enough clear and it is partly taken for granted. For readers not so familiarised with the literature, it would be good to state more clearly some reasons why measuring technological convergence is important, providing some examples.

In the revised version of the manuscript we discuss in detail the recent literature on COVID-19.

In particular, in the Introduction (Section 1) we followed the suggestion of reinforcing the reasons for which measuring technological convergence is important in the current situation. We discussed in particular the argument introduced by Lee and Trimi in the recent paper “Convergence innovation in the digital age and in the COVID-19 pandemic crisis.” (Journal of Business Research vol. 123 (2021): 14–22. doi:10.1016/j.jbusres.2020.09.041). We justify the need for rapid innovation and the reason for detecting rapidly technological convergence. Furthermore, we add a new Section (2) called “Innovation, Technology and coronavirus: state of the art” where we provide a literature review of recent developments concerning innovation and covid19.

Lastly, there are some concepts in the introductory section (like "resistance" or "resilience") whose meaning is not very precise. I would suggest reformulating them.

Thank you for the suggestion. In the revised manuscript we reformulate these concepts.

Reviewer #2: 

Report for “Rapid detection of fast innovation under the pressure of COVID-19”

General comments:

This paper presents a novel methodology to perform a rapid detection of the fast technological convergence phenomenon that is occurring under the pressure of the Covid-19 pandemic. The technological convergence phenomenon has been modelled through a network-based approach, analysing the differences of networks computed during two time periods (pre and post COVID-19). The results led us to discuss the repurposing of technologies regarding “Remote Control”, “Remote Working”, “Health” and “Remote Learning”. The research methods are basically reasonable. Specifically, there are several suggestions to help the author improve the quality of the paper.

1. It is suggested that in the part of introduction or research contribution, if the relevant literature of covid19 reflects the marginal contribution of this study, the following papers on covid19 can be cited.

Shen H, Fu M, Pan H, et al. The Impact of the COVID-19 Pandemic on Firm Performance[J]. Emerging Markets Finance and Trade, 2020, 56(10): 2213-2230.

Fu, M., & Shen, H. (2020). COVID-19 and corporate performance in the energy industry. Energy RESEARCH LETTERS, 1(1): 12967. https://doi.org/10.46557/001c.12967

Qin X, Huang G, Shen H, et al. COVID-19 Pandemic and Firm-level Cash Holding—Moderating Effect of Goodwill and Goodwill Impairment[J]. Emerging Markets Finance and Trade, 2020, 56(10): 2243-2258.

We revised the manuscript discussing in detail the literature of covid19.

First, we introduced a paragraph in the Introduction (Section 1) that reinforces the reasons for which measuring technological convergence is so important in the current situation. In particular, we introduced the recent considerations of Lee (see Lee, Sang M., and Silvana Trimi. “Convergence innovation in the digital age and in the COVID-19 pandemic crisis.” Journal of Business Research vol. 123 (2021): 14–22. doi:10.1016/j.jbusres.2020.09.041) that discussed the importance of Convergence Innovation in the current COVID-19 pandemic crisis.

Then, we added a new Section (2) called “Innovation, Technology and coronavirus: state of the art”. In this new Section we analyze the recent literature discussing the potential type of innovation generated during this pandemic. Furthermore, we discuss in detail the effect of the pandemic on firms citing the suggested literature. 

2. The lack of literature review on covid19 leads to inadequate theoretical research contribution. The above literature on covid19 can be cited.

We discuss this point in the above comment where we specify that we added a new section on theoretical research contribution on covid19.

3. This paper does not introduce which media platforms the technology clustering comes from.

Thank you for the comment. The media platform used to perform the clustering is Gephi (see Bastian M., Heymann S., Jacomy M. (2009). “Gephi: an open source software for exploring and manipulating networks)”. We specified the platform at the end of the introductory part of Section 6 (page 11) adding the reference.

4. What network platform data are used to build the novelty index (Ni) and the convergence index (CI)?

In the experiment, we exported the graph data in datatables using Gephi. Then, these tables have been manipulated using python with the numpy, pandas and sci-kit learn libraries. This has been mentioned in the introductory part of Section 6 (page 11).

5. This paper only compares the first quarter of 2019 with the first quarter of 2020. Why not the first quarter of 2020 and the fourth quarter of 2019? Or is it not a comparison of 2016, 2017, 2018, 2019 and 2020?

The reviewed version of the manuscript takes into account different time periods starting from 2017 (see Figure 10). The statistical analysis has been performed considering all of these periods. However, we leave the detailed discussion only for the first quarter of 2019 in order to compare similar time frames (the first quarter of the two year), leaving other seasonal or annual sources of variability.

6. After obtaining Ni and Ci, can we use the difference in difference (did) method in Shen h, Fu m, pan h, et al. The impact of the coved-19 pandemicon firm performance [J]. Emerging Markets Finance and trade, 2020, 56 (10): 2213-2230? Otherwise, without controlling other influencing factors, how can the author judge that these influences must be brought by covid19 instead of other factors?

We thank the reviewer for this precious comment. It gives us the possibility to discuss an appropriate statistical analysis.

Although your suggestion is fascinating, the use of the difference-in-differences (DID) to demonstrate what are the influencing factors of the calculation of the two indexes is problematic in practice. In particular, in this experimental study it is not possible to isolate a “control group” versus a “treatment group”. Since the coronavirus pandemic affected the overall community, it is difficult to build a “control group” of non-affected articles, especially in Medium. The only possibility would be to construct a sample of papers of the same post-Covid authors and investigate their pre-Covid production. This would require, however, a massive disambiguation effort, made more complex by the large presence of authors for which we ignore the status in bibliometric databases. Although the suggestion is very interesting, we believe it would require a brand new paper.

However, we argued the influence of COVID-19 on the results using another approach.

First, we emphasize that the two indexes are not affected in their calculation by the number of articles produced. Then we observe the behaviour of the two indexes in a time series. The time series suggests a sudden increase in the two indexes. We perform a Chow Test to test the null hypothesis of the absence of a structural break in the period when the covid-19 occured. The rejection of this null hypothesis leads us to conclude that the most suitable factor that influences these indexes is the coronavirus pandemic. Other competing explanatory factors (say, an endogenous shift in technologies) do not have the same time window.

We added a detailed discussion of this statistical analysis at the end of Section 6 (see also Figure 10). Again thank you for pointing to this issue: we feel the paper has now a stronger argument.

---

## [Decision Letter · Decision Letter 1]

7 Dec 2020

Rapid detection of fast innovation under the pressure of COVID-19

PONE-D-20-21013R1

Dear Dr. Bonaccorsi,

We’re pleased to inform you that your manuscript has been judged scientifically suitable for publication and will be formally accepted for publication once it meets all outstanding technical requirements. In this regard, abstract should be revised, whereas high-quality figures should be provided.

Kind regards,

Stefan Cristian Gherghina, PhD. Habil.

Academic Editor

PLOS ONE

Additional Editor Comments (optional):

Reviewers' comments:

Reviewer's Responses to Questions

**Comments to the Author**

1. If the authors have adequately addressed your comments raised in a previous round of review and you feel that this manuscript is now acceptable for publication, you may indicate that here to bypass the “Comments to the Author” section, enter your conflict of interest statement in the “Confidential to Editor” section, and submit your "Accept" recommendation.

Reviewer #1: All comments have been addressed

Reviewer #3: (No Response)

2. Is the manuscript technically sound, and do the data support the conclusions?

Reviewer #1: Yes

Reviewer #3: Yes

3. Has the statistical analysis been performed appropriately and rigorously? 

Reviewer #1: Yes

Reviewer #3: No

4. Have the authors made all data underlying the findings in their manuscript fully available?

Reviewer #1: Yes

Reviewer #3: Yes

5. Is the manuscript presented in an intelligible fashion and written in standard English?

Reviewer #1: Yes

Reviewer #3: Yes

6. Review Comments to the Author

Reviewer #1: All my comments has been addressed by the authors in a convincing way. In my previous review, I only raised several minor issues, do, once they have been addressed, I think that the manuscript can be accepted for publication.

Reviewer #3: Figures 1 and 10 have to be of very good quality and the caption should be detailed. The abstract must include a few sentences on the limitations as described in the conclusion section of the manuscript

7. PLOS authors have the option to publish the peer review history of their article (what does this mean?). If published, this will include your full peer review and any attached files.

Reviewer #1: **Yes: **José-Ignacio Antón

Reviewer #3: No

---

## [Editor Report · Acceptance letter]

15 Dec 2020

PONE-D-20-21013R1 

Rapid detection of fast innovation under the pressure of COVID-19 

Dear Dr. Bonaccorsi:

I'm pleased to inform you that your manuscript has been deemed suitable for publication in PLOS ONE. Congratulations! Your manuscript is now with our production department. 

Kind regards, 

on behalf of

Dr. Stefan Cristian Gherghina 

Academic Editor

PLOS ONE